# Remaining Useful Life Prediction of Cutting Tools Using an Inverse Gaussian Process Model

Yuanxing Huang [1], Zhiyuan Lu [2,*], Wei Dai [3], Weifang Zhang [3] and Bin Wang [4]

[1] School of Energy and Power Engineering, Beihang University, Beijing 100191, China; linhaihyx@126.com
[2] Beijing Hangxing Machinery Manufacturing Co., Ltd., Beijing 100013, China
[3] School of Reliability and Systems Engineering, Beihang University, Beijing 100191, China; dw@buaa.edu.cn (W.D.); zhangweifang@buaa.edu.cn (W.Z.)
[4] Beijing Spacecraft Co., Ltd., Beijing 100094, China; wb916@163.com
* Correspondence: rselzy@163.com

**Abstract:** In manufacturing, cutting tools gradually wear out during the cutting process and decrease in cutting precision. A cutting tool has to be replaced if its degradation exceeds a certain threshold, which is determined by the required cutting precision. To effectively schedule production and maintenance actions, it is vital to model the wear process of cutting tools and predict their remaining useful life (RUL). However, it is difficult to determine the RUL of cutting tools with cutting precision as a failure criterion, as cutting precision is not directly measurable. This paper proposed a RUL prediction method for a cutting tool, developed based on a degradation model, with the roughness of the cutting surface as a failure criterion. The surface roughness was linked to the wearing process of a cutting tool through a random threshold, and accounts for the impact of the dynamic working environment and variable materials of working pieces. The wear process is modeled using a random-effects inverse Gaussian (IG) process. The degradation rate is assumed to be unit-specific, considering the dynamic wear mechanism and a heterogeneous population. To adaptively update the model parameters for online RUL prediction, an expectation–maximization (EM) algorithm has been developed. The proposed method is illustrated using an example study. The experiments were performed on specimens of 7109 aluminum alloy by milling in the normalized state. The results reveal that the proposed method effectively evaluates the RUL of cutting tools according to the specified surface roughness, therefore improving cutting quality and efficiency.

**Keywords:** tool wear; remaining useful life; inverse Gaussian process; cutting precision; variable threshold

## 1. Introduction

Tool wear is widely considered to be stochastic and challenging to predict. This is primarily due to unit-to-unit performance variations and process variations. Efficient approaches that can predict remaining useful life (RUL) are necessary for improving cutting quality and saving costs. According to the input data used in the performance degradation model, RUL prediction methods can be classified into three categories: time series models (with working time as input), artificial intelligence models (with real-time working data as input), and stochastic process models (with degradation data as input).

The time series models analyze historical degeneration data to conduct RUL prediction using statistical approaches. Numerous time series models have been proposed and developed in tool RUL prediction in recent years, including the hidden Markov model [1,2], the autoregressive integrated moving average model [3], Kalman filtering [4,5], and particle filter [6–9]. Methods based on artificial intelligence take the extracted signal features or the original signal as input and RUL as output. Many advanced artificial intelligence models have been widely used in tool residual life prediction, such as neural network [10–12], support vector machines [13], and deep learning methods [14,15]. In stochastic process

models, degradation over time is often modeled by a stochastic process $\{Y(t); t \geq 0\}$ to account for damage accumulation, with inherent randomness. The RUL is determined as the first passage time of the process with respect to some failure threshold. In this context, three types of degradation modelling technique are widely discussed in the literature, namely the Wiener process model [16–18], the gamma process model [19], and the inverse Gaussian process model [20,21]. Meanwhile, Pimenov and Mikołajczyk combined neural networks and image processing for tool life prediction [22].

The time series models are suitable for mass production, with abundant historical degeneration data, while the artificial intelligence methods are appropriate for dealing with massive and complicated process data. Both the time series methods and the artificial intelligence methods are based on the invariable degradation trajectory. However, the tool wear process is complicated by randomness and periodicity, which are related to friction speed, pressure, surface roughness, material properties, friction and wear types, lubrication status, surface coating, and individual differences between tools. All of these factors lead to uncertainty in the tool degradation process. Therefore, it is more reasonable to describe tool performance degradation using stochastic processes. Although tools of the same type have commonalities in design and material, there might be significant individual differences due to dynamic use conditions. To characterize individual differences, the random-effects model is introduced in the stochastic process model [23]. Lu and Meeker [24] introduced a random variable into the degradation model to describe individual differences. Peng and Tseng [25] imposed a random effect on the drift parameter of the Wiener process, where a normal distribution is assumed for the random drift across the population. Compared to the Wiener process model and the gamma process model, the IG process is flexible in incorporating random effects that account for heterogeneities commonly observed in degradation problems [26,27].

In the current cutting process study, precision is generally used to describe the precision level of machine tool [28]. In the actual cutting process, the precision of the product is not only related to the precision of the machine, but also related to the level of the operator, the state of the cutting tool, the state of the fixture, the material characteristics of the cutting workpiece, and the processing technology. The whole machining system will affect the precision. In this paper, the research object is the cutting tool. Thus, the precision here refers to the precision of the cutting process. The influence of the cutting tool on cutting precision is directly reflected on the surface of a workpiece, which mainly has a significant influence on surface roughness. Therefore, the roughness of cutting surface is regarded as the index of cutting precision in this study. The time that the tool can normally work while still meeting the surface roughness requirements is defined as the RUL.

Compared with the wear of a tool, cutting precision criteria, such as the surface roughness, are more concerned with the actual cutting process [29]. Therefore, the failure criterion of the tool is generally not a decrease in its strength or stiffness but a decrease in its cutting precision. Traditional tool RUL prediction model focus on tool wear level. A conservative protection strategy could waste the RUL of the tool, increase unnecessary downtime and lead to a decrease in production efficiency. There are different mechanism models for the prediction of surface roughness have been studied [30–33]. The limitation of mechanism models is that it needs a lot of strict experiments test conditions, which is time-consuming and costly. In addition, the application of artificial intelligence technology in surface roughness has been widely discussed [34,35]. While these artificial intelligence models lack of discussion on tool wear degeneration, which is a very important factor related to surface quality. Thus, this paper proposes a dynamic evaluation method for RUL prediction, links the surface roughness to the wear of the tool, and the surface roughness criterion is modeled by a random threshold for the degradation state of the tool. The degradation of the wear process is modeled by an inverse Gaussian process, which has been successfully applied in degradation modeling [36]. Considering the quality variation of the tool, the degradation rate of the inverse Gaussian process is modelled as a random effect to improve the performance of the model.

The rest of this study is organized as follows: In Section 2, an inverse Gaussian process with a variable drift coefficient is formulated to characterize the degradation process considering the dynamic wear degradation mechanism and individual heterogeneity. The relationship between the surface roughness and degradation in terms of wearing is defined, and the RUL evaluation model with a random failure threshold is proposed. The parameter estimation procedure based on an EM algorithm is also developed. Section 3 provides the implementation and validation of the proposed approach through simulation experiments and real-data examples. The conclusions of the paper are drawn in Section 4.

## 2. Methods

### 2.1. Performance Degradation Modeling Based on Inverse Gaussian Process

In this paper, tool degradation is assumed to follow an inverse Gaussian process, as follows:

$$y(t) \sim \text{IG}(\mu \Lambda(t), \lambda[\Lambda(t)]^2, \tag{1}$$

where $\mu$ is related to degeneration rate, $\lambda$ presents the fluctuation of the degradation process, and $\Lambda(t)$ is a monotone increasing function. If the $\mu$ and $\lambda$ are known, the tool degradation process $Y(t)$ has independent increments. In addition, $Y(t)$ is also monotonically increasing. For a given failure threshold $\omega$, the failure time T of the tool can be defined as the first passage of time when $Y(t)$ exceeds the threshold $\omega$. Accordingly, the probability density function (PDF) and cumulative distribution function (CDF) of failure time T can be obtained as follows [37]:

$$F_{\text{T}}(t) = \text{P}(t < t) = \text{P}(y(t) > \omega)$$

$$= \Phi\left[\sqrt{\frac{\lambda}{\omega}}\left(\Lambda(t) - \frac{\omega}{\mu}\right)\right] - \exp\left(\frac{2\lambda\Lambda(t)}{\mu}\right)\Phi\left[-\frac{\lambda}{\omega}\left(\Lambda(t) + \frac{\omega}{\mu}\right)\right], \tag{2}$$

and

$$f_{\text{T}}(t) = \sqrt{\frac{\lambda}{\omega}}\Lambda'(t)\Phi\left[\sqrt{\frac{\lambda}{\omega}}\left(\Lambda(t) - \frac{\omega}{\mu}\right)\right] - \frac{2\lambda}{\mu}\Lambda'(t)\exp\left(\frac{2\lambda\Lambda(t)}{\mu}\right)\Phi\left[-\frac{\lambda}{\omega}\left(\Lambda(t) + \frac{\omega}{\mu u}\right)\right]$$

$$+ \sqrt{\frac{\lambda}{\omega}}\Lambda'(t)\exp\left(\frac{2\lambda\Lambda(t)}{\mu}\right)\Phi\left[-\frac{\lambda}{\omega}\left(\Lambda(t) + \frac{\omega}{\mu}\right)\right], \tag{3}$$

Due to the variability in the raw materials and the dynamic working conditions, the degradation rate $\mu$ itself can vary from unit to unit. We assume that the degradation rate $\mu$ can be modeled as a random effect, which follows a certain distribution to account for this aspect. The typical model for the random effects for $\mu$ in the IG process includes the truncated normal distribution and gamma distribution. In this study, we assume that $1/\mu$ follows a normal distribution $N\left(\alpha_\mu, \sigma_\mu^{-2}\right)$, considering the two parameters correspond to the mean and variance of degradation rate respectively. The model parameters of normal distribution have definite physical meaning, so it is convenient to quantify the subjective information such as expert information. Then according to the total probability formula, the CDF of residual life Tr, considering the random degradation rate, can be expressed as follows [38]:

$$F_{\text{Tr}}(t) = \int F_{\text{T}}(t|1/\mu)f_{1/\mu}(x)dx$$

$$= \text{E}_{1/\mu}\left[\Phi\left(\sqrt{\frac{\lambda}{\omega}}\left(\Lambda(t) - \frac{\omega}{\mu}\right)\right)\right] + \text{E}_{1/\mu}\left[\exp\left(\frac{2\lambda\Lambda(t)}{\mu}\right)\Phi\left(-\sqrt{\frac{\lambda}{\omega}}\left(\Lambda(t) + \frac{\omega}{\mu}\right)\right)\right], \tag{4}$$

In order to simplify the course of the derivation for the RUL distribution, two lemmas are given [16,39]:

**Lemma 1.** *If* $Y \sim N(\alpha_1, \sigma_1^2)$, *and* $a, b \in R$, *then the following holds*:

$$\mathrm{E}_Y[\Phi(a + bY)] = \Phi\left[(a + b\alpha_1) / \left(1 + b^2\sigma_1^2\right)\right], \tag{5}$$

**Lemma 2.** *If* $Z \sim N(\alpha_2, \sigma_2^2)$, *and* $A, B, C \in R$, *then the following holds*:

$$\mathrm{E}_Z[\exp(AZ)\Phi(B + CZ)] = \exp\left(A\alpha_2 + \frac{A^2\sigma_2^2}{2}\right)\Phi\left(\frac{B + C\alpha_2 + AC\sigma_2^2}{\sqrt{1 + C^2\sigma_2^2}}\right), \tag{6}$$

Based on Lemmas 1 and 2, we can calculate (4) explicitly. The CDF and PDF of residual life Tr can be formulated as:

$$F_{\mathrm{Tr}}(t) = \Phi\left(\sqrt{\frac{\lambda}{\omega}}\frac{\sigma_\mu\Lambda(t) - \alpha_\mu\sigma_\mu\omega}{\sqrt{\sigma_\mu^2 + \lambda\omega}}\right) - \exp\left(2\alpha_\mu\lambda\Lambda(t) + \frac{2\lambda^2\Lambda^2(t)}{\sigma_\mu^2}\right) \times \Phi\left[-\sqrt{\frac{\lambda}{\omega}}\frac{\left(\sigma_\mu^2 + 2\lambda\omega\right)\Lambda(t) + \alpha_\mu\sigma_\mu^2\omega}{\sqrt{\sigma_\mu^4 + \lambda\omega\sigma_\mu^2}}\right], \tag{7}$$

and

$$\begin{aligned}
f_{\mathrm{Tr}}(t) = {} & \sqrt{\frac{\lambda}{\omega}}\frac{\sigma_\mu}{\sqrt{\sigma_\mu^2 + \lambda\omega}}\Lambda'(t) \times \Phi\left[\sqrt{\frac{\lambda}{\omega}}\frac{\sigma_\mu\Lambda(t) - \alpha_\mu\sigma_\mu\omega}{\sqrt{\sigma_\mu^2 + \lambda\omega}}\right] - \exp\left(2\alpha_\mu\lambda\Lambda(t) + \frac{2\lambda^2\Lambda^2(t)}{\sigma_\mu^2}\right) \\
& \times \left(2\alpha_\mu\lambda\Lambda'(t) + \frac{2\lambda^2\Lambda^2(t)}{\sigma_\mu^2}\Lambda'(t)\right)\Phi\left[-\sqrt{\frac{\lambda}{\omega}}\frac{\left(\sigma_\mu^2 + 2\lambda\omega\right)\Lambda(t) + \alpha_\mu\sigma_\mu^2\omega}{\sqrt{\sigma_\mu^4 + \lambda\omega\sigma_\mu^2}}\right] \\
& + \sqrt{\frac{\lambda}{\omega}}\frac{\left(\sigma_\mu^2 + 2\lambda\omega\right)}{\sqrt{\sigma_\mu^4 + \lambda\omega\sigma_\mu^2}} \times \Lambda'(t)\exp\left(2\alpha_\mu\lambda\Lambda(t) + \frac{2\lambda^2\Lambda^2(t)}{\sigma_\mu^2}\right) \\
& \times \Phi\left[-\sqrt{\frac{\lambda}{\omega}}\frac{\left(\sigma_\mu^2 + 2\lambda\omega\right)\Lambda(t) + \alpha_\mu\sigma_\mu^2\omega}{\sqrt{\sigma_\mu^4 + \lambda\omega\sigma_\mu^2}}\right].
\end{aligned} \tag{8}$$

### 2.2. Remaining Useful Life Evaluation Model

In the actual cutting process, cutting precision is not only related to the precision of the machine tool but also to the level of the operator, the state of the cutting tool, the state of the fixture, the material characteristics of the cutting workpiece, and the processing technology. However, these factors in the manufacturing process are often steady, while the cutting tool is gradually worn. Therefore, it is often the case that cutting precision is mainly dependent on the wearing level of the cutting tool.

The influence of the cutting tool on cutting precision is directly reflected on the surface of a workpiece, which mainly has a significant influence on surface roughness. Geometric dimension precision is more related to the whole machining system, which can be improved by the adjustment of the operator. In addition, the mechanism affecting dimensional precision, such as the performance degradation of the spindle, will not change significantly in a short time. However, the tool wear will change significantly in a relatively short working time, resulting in abnormal surface roughness. Therefore, in this study, surface roughness is considered as the most significant precision index caused by tool wear in the short term.

The time that the tool can normally work while still meeting the surface roughness requirements is defined as the remaining useful life (RUL). Assume that $R_k$ is the RUL corresponding to the equipment at the current measurement time $t_k$; that is, the interval from time $t_k$ to the time of fault occurrence.

$$R_k = \inf\{r_k : y(t_k + r_k) > \omega | \mathbf{Y}_{0:k}\}, \tag{9}$$

where $\mathbf{Y}_{0:k}$ is the degenerate historical dataset from start time $t_0$ to time $t_k$.

In addition, function $\Lambda(t)$ should be updated under a different measurement time. The function $\Lambda(t)$ corresponding to the measurement time $t_k$ is $\Lambda^{(t_k)}(t)$:

$$\Lambda^{(t_k)}(t) = \Lambda^{(0)}(t + t_k) - \Lambda^{(0)}(t),\tag{10}$$

where $\Lambda^{(0)}(t)$ is the function $\Lambda(t)$ at the initial time.

According to Equations (7)–(10), the CDF and PDF corresponding to remaining useful life at $t_k$ are as follows:

$$F_{R_k}(r_k) = \Phi\left(\sqrt{\frac{\lambda}{\omega - y_k}}\frac{\sigma_\mu \Lambda^{(t_k)}(r_k) - \alpha_\mu \sigma_\mu(\omega - y_k)}{\sqrt{\sigma_\mu^2 + \lambda(\omega - y_k)}}\right) - \exp\left(2\alpha_\mu\lambda\Lambda^{t_k}(r_k) + \frac{2\lambda^2\Lambda^{(t_k)2}(r_k)}{\sigma_\mu^2}\right)$$

$$\times\Phi\left[-\sqrt{\frac{\lambda}{\omega + y_k}}\frac{\left(\sigma_\mu^2 + 2\lambda(\omega - y_k)\right)\Lambda^{(t_k)}(r_k) + \alpha_\mu\sigma_\mu^2(\omega - y_k)}{\sqrt{\sigma_\mu^4 + \lambda(\omega - y_k)\sigma_\mu^2}}\right],\tag{11}$$

and

$$f_{R_k}(r_k) = \sqrt{\frac{\lambda}{\omega - y_k}}\frac{\sigma_\mu}{\sqrt{\sigma_\mu^2 + \lambda(\omega - y_k)}} \times \Phi\left[\sqrt{\frac{\lambda}{\omega - y_k}}\frac{\sigma_\mu r_k - \alpha_\mu\sigma_\mu(\omega - y_k)}{\sqrt{\sigma_\mu^2 + \lambda(\omega - y_k)}}\right]$$

$$- \exp\left(2\alpha_\mu\lambda r_k + \frac{2\lambda^2 r_k^2}{\sigma_\mu^2}\right) \times \left(2\alpha_\mu\lambda r_k + \frac{2\lambda^2 r_k^2}{\sigma_\mu^2}\right)$$

$$\times\Phi\left[-\sqrt{\frac{\lambda}{\omega - y_k}}\frac{\left(\sigma_\mu^2 + 2\lambda(\omega - y_k)\right)r_k + \alpha_\mu\sigma_\mu^2(\omega - y_k)}{\sqrt{\sigma_\mu^4 + \lambda(\omega - y_k)\sigma_\mu^2}}\right]$$

$$+ \sqrt{\frac{\lambda}{\omega - y_k}}\frac{\sigma_\mu^2 + 2\lambda(\omega - y_k)}{\sqrt{\sigma_\mu^4 + \lambda(\omega - y_k)\sigma_\mu^2}} \times \exp\left(2\alpha_\mu\lambda r_k + \frac{2\lambda^2 r_k^2}{\sigma_\mu^2}\right)$$

$$\times\Phi\left[-\sqrt{\frac{\lambda}{\omega - y_k}}\frac{\left(\sigma_\mu^2 + 2\lambda(\omega - y_k)\right)r_k + \alpha_\mu\sigma_\mu^2(\omega - y_k)}{\sqrt{\sigma_\mu^4 + \lambda(\omega - y_k)\sigma_\mu^2}}\right],\tag{12}$$

It is difficult or even impossible to predetermine a failure threshold in many scenarios. One possible method to tackle the above-mentioned problem is to assume the failure threshold follows a specified distribution [40,41]. For a given surface roughness requirement, the wear failure threshold $\omega$ for the wearing process is a random variable in a change interval of $[\omega_L, \omega_U]$. In this study, the $\omega$ is assumed to obey uniform distribution. Then, the CDF of the remaining useful life can be expressed as follows:

$$F_{R_k}(r_k|Ra) = \frac{1}{\omega_U - \omega_L}\int_{\omega_L}^{\omega_U}F_{R_k}(r_k)d\omega,\tag{13}$$

The average remaining useful life can be calculated as follows:

$$T_{r_k} = \int_0^\infty\left(1 - F_{R_k}(r_k|Ra)\right)dr_k.\tag{14}$$

The method to estimate parameters $\theta = \left(\alpha_\mu, \sigma_\mu^2, \lambda\right)$ in the above model will be introduced in Section 2.3.

### 2.3. Parameter Estimation Based on Expectation–Maximization (EM)

As the random variable parameter $1/\mu$ cannot be observed directly, an expectation–maximum (EM) algorithm is applied to estimate its value. The EM algorithm is an iterative optimization strategy that includes an E-step and an M-step in an iteration. In each iteration, the conditional distribution of the missing data and the expectation of the complete log-likelihood, with respect to the conditional distribution of the missing data, are derived in the E-step, with the model parameters estimated in the previous step. The estimates for model parameters are then updated by maximizing the expectation of the complete log-likelihood in the M-step. The iteration is repeated until the estimates for model parameters converge.

Since we have assumed that $1/\mu \sim N\left(\alpha_\mu, \sigma_\mu^{-2}\right)$, according to the Bayesian formula, the posterior distribution of $1/\mu_k$ at $t_k$ can be obtained according to the Bayesian formula:

$$p\left(\frac{1}{\mu_k}\middle|\mathbf{Y}_{0:k}\right) \propto p\left(\mathbf{Y}_{0:k}\middle|\frac{1}{\mu_k}\right)p\left(\frac{1}{\mu_k}\right) \propto \prod_{i=1}^{k}\exp\left[-\frac{\lambda_k(\Delta y_i - \mu_k\Delta\Lambda(t_i))^2}{2\mu_k\Delta y_i}\right]\exp\left[-\frac{\sigma_{\mu,k^2}\left(1 - \mu_k\alpha_{\mu,k}\right)^2}{2\mu_k^2}\right]$$

$$= \exp\left\{-\frac{\lambda_k}{2\mu_k^2}\sum_{i=1}^{k}\frac{(\Delta y_i - \mu_k\Delta\Lambda(t_i))^2}{\Delta y_i} - \frac{\sigma_{\mu,k^2}\left(1 - \mu_k\alpha_{\mu,k}\right)^2}{2\mu_k^2}\right\}$$

$$\propto \exp\left[-\frac{\lambda_k y_k + \sigma_{\mu,k^2}}{2}\left(\frac{1}{\mu_k^2} - 2\frac{\lambda_k\Lambda(t_k) + \alpha_{\mu,k}\sigma_{\mu,k^2}}{\mu_k\left(\lambda_k y_k + \sigma_{\mu,k^2}\right)}\right)\right] \sim N\left(\alpha_{0,k}, \sigma_{0,k}^{-2}\right), \tag{15}$$

where

$$\alpha_{0,k} = \frac{\lambda_k\Lambda(t_k) + \alpha_{\mu,k}\sigma_{\mu,k}^2}{\lambda_k y_k + \sigma_{\mu,k}^2}, \tag{16}$$

$$\sigma_{0,k}^2 = \lambda_k y_k + \sigma_{\mu,k}^2, \tag{17}$$

Define $\theta_k^j = \left(\alpha_{u,k}^j, \sigma_{u,k}^{2(j)}, \lambda_{u,k}^j\right)$ as the parameter θ at time $t_k$, where $j$ is iteration times. The complete log-likelihood function of $\{\mathbf{Y}_{0:k}, \frac{1}{\mu}\}$ can be expressed as follows:

$$L\left(\theta\middle|\mathbf{Y}_{0:k}, \frac{1}{\mu}\right) = \ln\left[p\left(\mathbf{Y}_{0:k}, \frac{1}{\mu}\middle|\theta\right)\right] = \ln\left[p\left(\mathbf{Y}_{0:k}\middle|\frac{1}{\mu}, \theta\right)p\left(\frac{1}{\mu}\middle|\theta\right)\right]$$

$$= \frac{k}{2}\ln(\lambda) - \frac{k+1}{2}\ln 2\pi + \sum_{i=1}^{k}\ln(\Delta\Lambda(t_i)) - \frac{3}{2}\sum_{i=1}^{k}\ln(\Delta y_i) + \frac{1}{2}\ln(\sigma_\mu^2)$$

$$- \sum_{i}^{k}\frac{\lambda(\Delta y_i - \mu\Delta\Lambda(t_i))^2}{2\mu^2\Delta y_i} - \frac{\sigma_\mu^2}{2}\left(\frac{1}{\mu} - \alpha_\mu\right)^2, \tag{18}$$

By maximizing the complete log-likelihood function, the parameter θ is calculated as follows:

$$\alpha_{\mu,k}^{j+1} = E\left(\frac{1}{\mu}\right), \tag{19}$$

$$\sigma_{\mu,k}^{2(j+1)} = D\left(\frac{1}{\mu}\right)^{-1}, \tag{20}$$

$$\lambda_\mu^{j+1} = \frac{k}{E\left(\frac{1}{\mu^2}\right)y_k - 2E\left(\frac{1}{\mu}\right)\Lambda(t_k) + \sum_{i=1}^{k}\frac{\Delta\Lambda^2(t_i)}{\Delta y_i}}. \tag{21}$$

where $E\left(\frac{1}{\mu}\right)$ and $D\left(\frac{1}{\mu}\right)^{-1}$ are the expectation and variance of $\frac{1}{\mu}$ conditional on $\mathbf{Y}_{0:k}$ given in (16) and (17). The optimal parameters $\hat{\theta}_k = \theta_k^{j+1}$ can be obtained by the iteration until

algorithm convergence. A large portion of the existing literature on the EM algorithm has proven that the algorithm is not only simple in calculation but can also guarantee convergence. With the increase in iteration times, the likelihood function will also increase, so that the result will improve under the maximum likelihood function. Because $Y_{0:k}$ is obtained with the continuous measurement of processing, the algorithm can be used to estimate the model parameters at any time after obtaining the degradation data. Furthermore, with the increase in available data, the estimated model parameters will be more accurate.

## 3. Example Study

### 3.1. Simulation

For validating the performance of the algorithm of the proposed approach, the following simulation was conducted. The 100 numbered simulated data were generated, under the assumption that the parameters in the inverse Gaussian model were set as $\alpha\mu = 1$, $\sigma\mu = 100$, $\lambda = 2$. The data information, including simulated degradation data, was recorded with the cycle. The degradation trajectory obtained by the simulation is shown in Figure 1. The former 10 data points were regarded as historical data. The parameters $\alpha\mu$, $\sigma\mu$, and $\lambda$ were estimated at different times by the EM algorithm. The simulated data contributed to the performance analysis of the proposed model, without physical meaning. Figures 2–4 show that the estimated values of parameters are approaching the real values, with the accumulation of simulated degradation data.

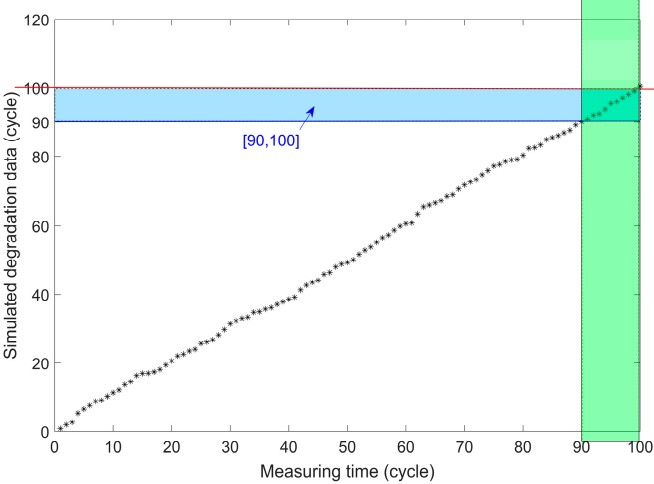

**Figure 1.** Degradation data from simulation.

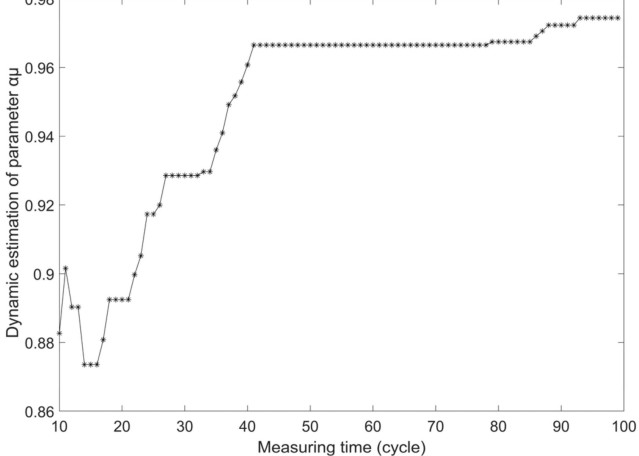

**Figure 2.** Dynamic estimation of parameter $\alpha\mu$.

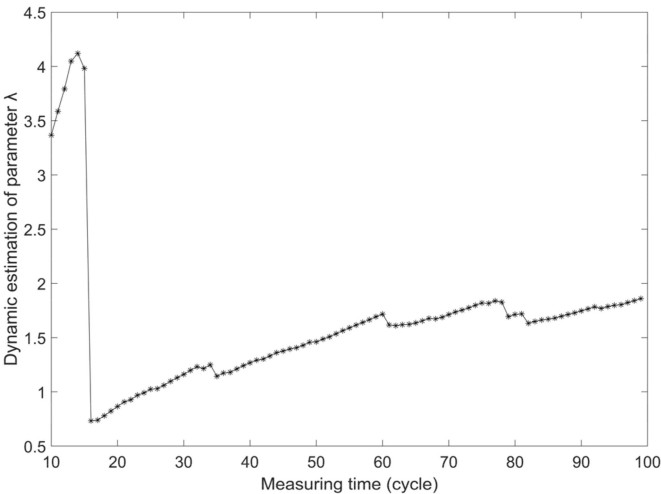

**Figure 3.** Dynamic estimation of parameter $\lambda$.

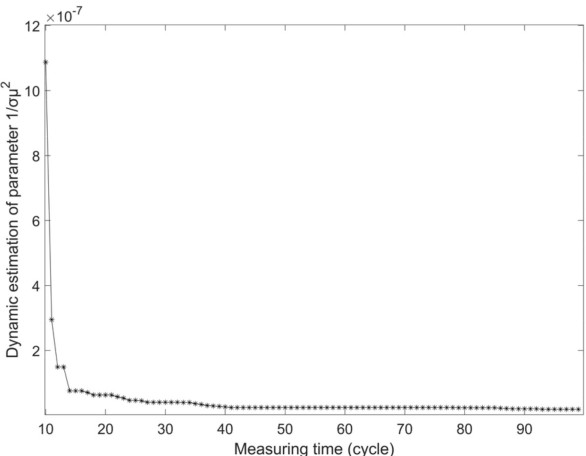

**Figure 4.** Dynamic estimation of parameter $\sigma\mu$.

From Figure 5, we can see that, with more and more data available, the PDF of RUL will be narrower, which indicates that the uncertainty of the prediction results becomes smaller, and the corresponding point estimation is closer to the real RUL. In this simulation, the failure threshold was set to [90, 100], and it obeyed uniform probability distribution. Subsequently, the prediction result is drawn in Figure 6.

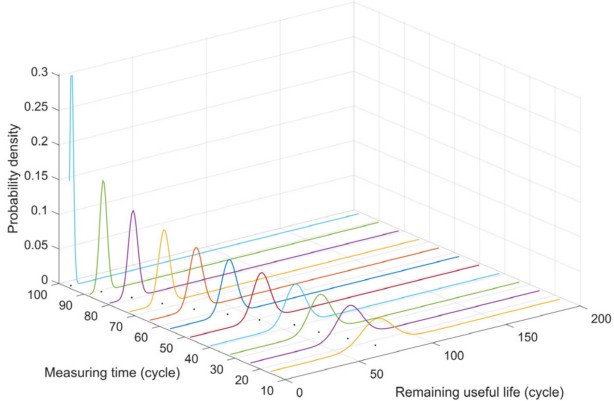

**Figure 5.** PDF of RUL with simulation data.

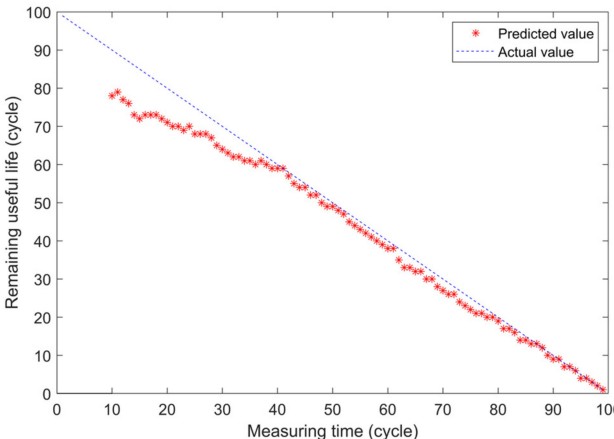

**Figure 6.** Predicted value and actual value of URL with simulation data.

### 3.2. Experiment

Real milling experiments were performed to verify the availability and validity of the proposed approach by detecting tool wear condition. The experiment was performed on specimens of 7109 aluminum alloy by milling in the normalized state. The workpiece was cuboid with a 100-mm side length. The chemical compositions of the selected materials are specified in Table 1.

**Table 1.** Chemical compositions of material of the cutting workpiece.

| Material | Chemical Composition % | | | | | | | | | | |
|---|---|---|---|---|---|---|---|---|---|---|---|
| | Si | Fe | Cu | Mn | Mg | Cr | Zn | Zr | Co | O | Ti |
| 7109 | 0.1 | 0.15 | 0.1 | 0.05 | 2.4 | 0.06 | 6.2 | 0.15 | 0.2 | 0.05 | 0.1 |

The experiment was carried out using a flat-end milling cutter using coolant, and the characteristics of the tool are specified in Table 2. A five-axis DMG CTX gamma 2000TC (Hamburg, Germany) with Numeric Control Siemens 840D sl (Munich, Germany) was used in the experiment. The cutting was conducted with the spindle speed of the cutter at 6000 r/min, a feed rate value of 2000 mm/min, and a cutting depth value of 1 mm. The cutter wear was measured using a Dino-Lite AM3113 microscopy system (AnMo, Shenzhen, China). The roughness of the machined surface was measured by the Mahr M1 surface roughness meter (Mahr GmbH, Esslingen am Neckar, Germany) after the cutting. The proposed approach was run on a server with a 2.40 GHz processor and 64 GB RAM.

**Table 2.** Characteristics of the tool.

| Grade | Helical Angle | Number of Teeth | Diameter | Over Length | Edge Length | Cutting Edge Diameter | Material |
|---|---|---|---|---|---|---|---|
| ZCC.CT (China) | 55° | 3 | 6 mm | 50 mm | 12 mm | 4 mm | Cemented Carbide |

Milling from the lower edge of the workpiece to the upper edge of the workpiece was recorded as a cycle. After each cutting cycle, we stopped and collected the roughness data of the specimens. The roughness was measured on the flank face of the cutting surface four times, and the four measurements were averaged as the true roughness. Meanwhile, the tool wear was measured using a Dino-Lite microscopy system. The wear of tool and the roughness of workpiece were monitored and recorded, until the roughness deviated from the requirement value of 2 μm. The experimental environments and measurements are shown in Figure 7.

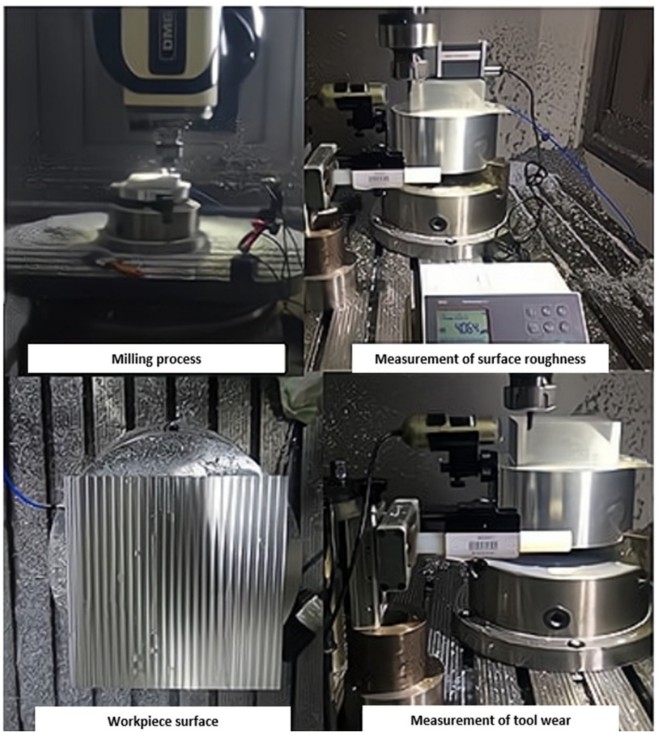

**Figure 7.** Experiment environment and measurement.

Due to equipment and artificial measurement errors, there may be fluctuation errors in wear measurement results. To eliminate measurement errors, abnormal data were rejected, and the mean values of the normal data before and after were filled in. Then, the measurement data were smoothed. The tool-wearing curve is shown in Figure 8. Considering the high rate in the early stage and the severe wear in the later stage, the data of a stable wear period were applied for the modeling.

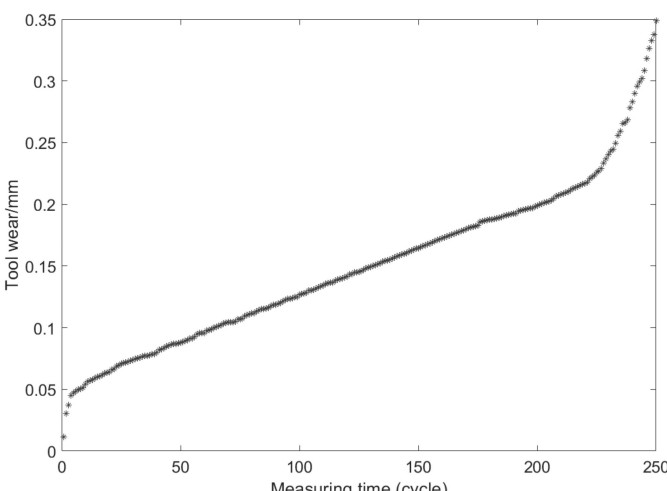

**Figure 8.** Tool-wearing curve.

In this experiment, it was assumed that the failure criterion of the tool is whether the surface roughness of the workpiece met 2 μm. The former 40 cycles of cutting use were regarded as the historical data. The prediction results were not ideal, with a fixed failure threshold of 0.18 mm, as shown in Figure 9. Then, the failure threshold was set to [0.175mm, 0.18mm] and obeyed a uniform probability distribution. As shown in Figure 10,

the prediction results improved significantly compared to Figure 3. The PDF of the URL was drawn as shown in Figure 11.

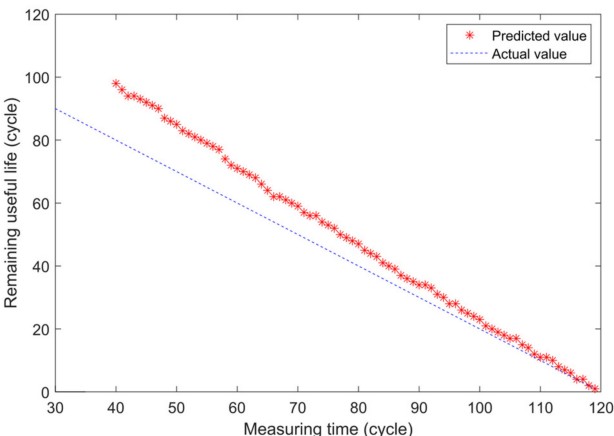

**Figure 9.** Predicted value and actual value of URL with fixed degradation threshold.

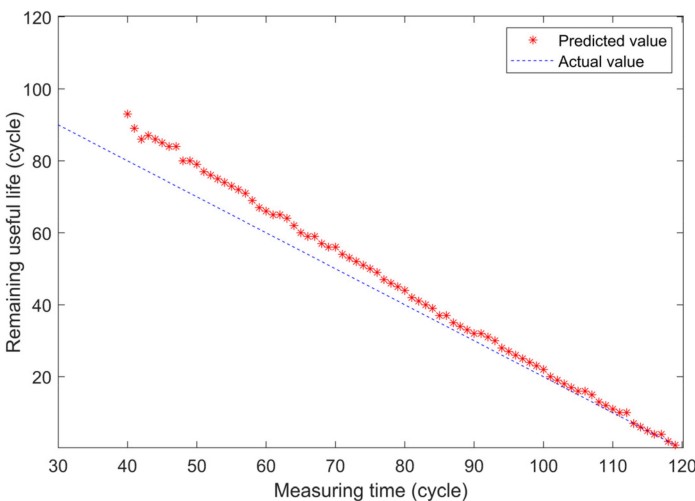

**Figure 10.** Predicted value and actual value of URL with variable degradation threshold.

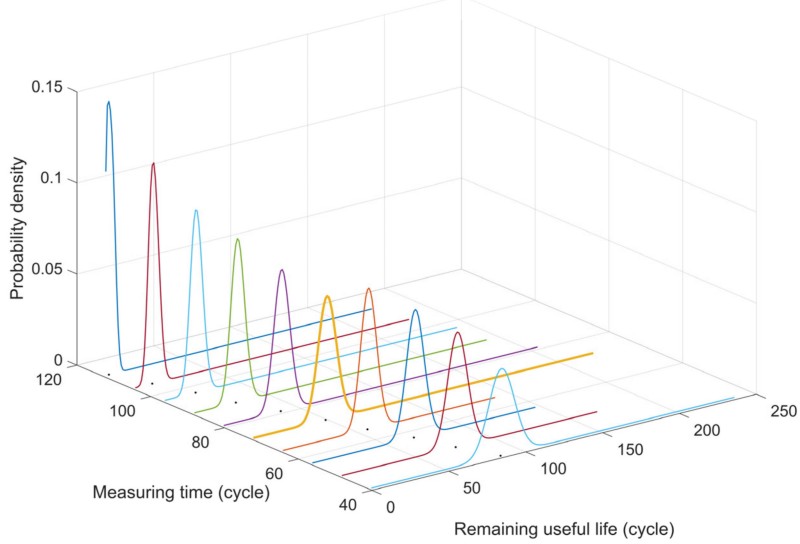

**Figure 11.** PDF of URL and its corresponding point estimation under variable degradation threshold.

### 3.3. Comparison of the RUL Predictive Model

Based on the estimated RUL, the mean absolute error (MAE) between the estimated RUL and the true RUL can be calculated as follows:

$$\text{MAE} = \frac{1}{N} \sum_{i=1}^{N} \left( \left| \text{RUL}_{\text{predicted}}^{i} - \text{RUL}_{\text{actual}}^{i} \right| \right). \tag{22}$$

Consequently, the MAE from the 40th cycle to the 120th cycle can be used as a measure to quantify the prediction accuracy of the model. Figure 12 presents the MAE of the estimated RUL using the model with a variable degradation threshold and the model with a fixed degradation threshold. Apparently, the model with a variable degradation threshold gives a more precise RUL prediction.

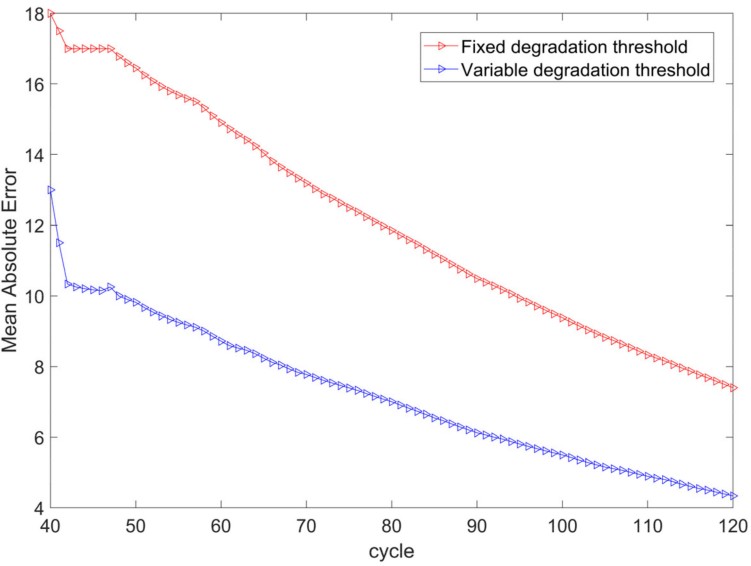

**Figure 12.** MAE between the estimated RUL and the true RUL.

In order to verify the effect of the model, a comparison was made between the IG process model and a particle filtering method using deviation accuracy [8]. The deviation accuracy was set as 0.1, which means that the prediction distribution falls within $1 \pm 0.1$ of true RUL and is regarded as the performance indicator of the two models. Table 3 shows the result of the comparison, which demonstrates the better performance of the method proposed in this paper.

**Table 3.** Distributions of predicted RUL based on deviation accuracy.

| Model | Prediction Distribution Falls within $1 \pm 0.1$ of True RUL |
|---|---|
| IG process model | 72% |
| Particle filtering model | 60% |

## 4. Conclusions

In this paper, we studied the degradation modeling and RUL prediction of cutting tools based on a cutting precision criterion. The cutting tool suffers continued wear in usage, which decreases the cutting precision of the machining process, i.e., the roughness of the surface in this study. Although cutting precision is of more practical value, it is indirectly measurable and its degradation pattern is more complex. On the other hand, the wear state of a tool is a more directly measurable characteristic in practice, and its degradation pattern is more traceable. Therefore, we proposed to model the degradation process of tool wear as a proxy for the degradation of cutting precision, and linked the

roughness of the surface requirement to a random threshold for the wearing of a tool. A degradation model based on the IG process was proposed for the tool wearing process, and the RUL prediction method was also studied. The following conclusive remarks were reached in this study:

1.  An IG process model with a variable drift coefficient was used to characterize the degradation of the tool wearing process subjected to individual heterogeneity in dynamic working environments;
2.  The surface roughness requirement was linked to a random threshold for the wearing of the cutting tool, and the RUL prediction method was developed based on the proposed degradation model with a random failure threshold.
3.  Finally, the applicability and effectiveness of the proposed method was validated using the wearing data of cutting tools in a milling experiment; the MAE was 4.33.

Further work is required to extend the proposed model's generalizability for handling the multiple cutting conditions observed in real cutting processes, such as turning, planing, and grinding. In addition, the distribution of the failure variable threshold is subject to confirmation by experimental and statistical analyses.

**Author Contributions:** Y.H.: Conceptualization, methodology, software, and writing—original draft. Z.L.: data curation, validation, and investigation. W.D. and W.Z.: supervision and writing—review and editing. B.W.: writing—review and editing. All authors have read and agreed to the published version of the manuscript.

**Funding:** This research was funded by National Fundamental Research Foundation of China (No. JSZL2019601A003 and YWF-21-BJ-J-727) and Technical foundation program of the Ministry of Industry and Information Technology of China (No. JSZL2019601C006).

**Conflicts of Interest:** The authors declare no conflict of interest. The funders had no role in the design of the study; in the collection, analyses, or interpretation of data; in the writing of the manuscript, or in the decision to publish the results.

## Abbreviation

| | |
|---|---|
| RUL | remaining useful life |
| IG | inverse Gaussian |
| EM | expectation-maximization |
| PDF | probability density function |
| CDF | cumulative distribution function |
| Ra | surface roughness |
| $Y(t)$ | degradation process with a simple IG process model |
| $\mu$ | degeneration rate of $Y(t)$ |
| $\lambda$ | fluctuation coefficient of $Y(t)$ |
| $\Lambda(t)$ | monotone increasing function of $Y(t)$ |
| $\omega$ | failure threshold |
| T | failure time |
| Tr | residual life |
| $F_T(t)$ | CDF of T |
| $f_T(t)$ | PDF of T |
| $P(\cdot)$ | probability of an event |
| $E(\cdot)$ | expectation operator |
| $N(a, b)$ | uniform distribution with boundary [a, b] |
| $\Phi(\cdot)$ | CDF of standard normal distribution |
| $N\left(\alpha_\mu, \sigma_\mu^{-2}\right)$ | distribution of Parameter $1/\mu$ |
| $\Lambda'(t)$ | derivative function of $\Lambda(t)$ |
| $t_k$ | $k$th measurement time |
| $R_k$ | RUL corresponding to the equipment at the current measurement time $t_k$ |

| | |
|---|---|
| $\mathbf{Y}_{0:k}$ | historical degenerate dataset from start time $t_0$ to time $t_k$. |
| $\Lambda^{(t_k)}(t)$ | $\Lambda(t)$ at the measurement time $t_k$ |
| $\Lambda^{(0)}(t)$ | $\Lambda(t)$ at the initial time |
| $F_{\mathrm{Tr}}(t)$ | CDF of Tr |
| $f_{\mathrm{Tr}}(t)$ | PDF of Tr |
| $F_{\mathrm{Rk}}(r_k)$ | CDF of $R_k$ at $tk$ |
| $f_{\mathrm{Rk}}(r_k)$ | PDF of $R_k$ at $tk$ |
| $y_k$ | degradation value at time $t_k$ |
| $\Delta y$ | degradation increment |
| $j$ | iteration times |
| $\theta$ | estimated parameters $\theta = (\alpha\mu, \sigma\mu^{-2}, \lambda)$ |
| $\theta_k^j$ | $\theta_k^j = \left( \alpha_{u,k}^j, \sigma_{u,k}^{2(j)}, \lambda_{u,k}^j \right)$ is the parameter $\theta$ at time $t_k$ after $j$ iterations |
| $p\left( \frac{1}{\mu_k} \middle| \mathbf{Y}_{0:k} \right)$ | posterior distribution of $1/\mu_k$ at $t_k$ |
| $p\left( \mathbf{Y}_{0:k} \middle| \frac{1}{\mu_k} \right)$ | joint log-likelihood function for observed events, $\mathbf{Y}_{0:k}$ and $1/\mu_k$ |
| $p\left( \frac{1}{\mu_k} \right)$ | prior distribution of $1/\mu_k$ at $t_k$ |
| $L\left( \theta \middle| \mathbf{Y}_{0:k}, \frac{1}{\mu} \right)$ | complete log-likelihood function of $\{\mathbf{Y}_{0:k}, \frac{1}{\mu}\}$ |
| $p\left( \mathbf{Y}_{0:k}, \frac{1}{\mu} \middle| \theta \right)$ | joint density function for observed events, $\mathbf{Y}_{0:k}$, $1/\mu$, and $\theta$ |
| $p\left( \mathbf{Y}_{0:k} \middle| \frac{1}{\mu}, \theta \right)$ | conditional probability density, with the parameter $1/\mu$ and parameter $\theta$ are known |
| $p\left( \frac{1}{\mu} \middle| \theta \right)$ | joint density function for observed events,$1/\mu$ and $\theta$ |
| $\hat{\theta}_k$ | optimal parameters |
| MAE | mean absolute error |
| $RUL_{\mathrm{predicted}}^i$ | predicted RUL at the $i$ cycle |
| $RUL_{\mathrm{actual}}^i$ | actual RUL at the $i$ cycle |

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
