# Peer review of "Remaining Useful Life Prediction of Cutting Tools Using an Inverse Gaussian Process Model"

_applsci, doi:10.3390/app11115011_

Round 1
Reviewer 1 Report
The authors proposed a dynamic RUL prediction method for cutting tools based on their degradation process. However, the method does not have significant novelty, the manuscript is organized well. The manuscript can be considered for publication after consideration of the below comments.
1- The authors should explain why the wear process is modeled by a random-effects inverse Gaussian (IG) process? What is the advantage of this model?
2- Introduction needs to improve using the below citations. They need to discuss more other methods and models in this domine.
-Tool wear prediction in high-speed turning of a steel alloy using long short-term memory modelling." Measurement 177 (2021): 109329.
-Prediction of cutting tool wear during a turning process using artificial intelligence techniques." The International Journal of Advanced Manufacturing Technology 111.1 (2020): 505-515.
Reviewer 2 Report
The prediction of the remaining useful life (RUL) of the cutting tools is important. If the moment of the tool change is delayed, the dimensional precision and the quality of the machined workpieces can be affected.
In their attempt to precisely predict the RUL of tools, the authors combine two important parameters: the roughness of the machined surface and the wear threshold of the tool.
Even the paper is interesting, there are many aspects that must be clarified before recommending it for publication. To this end, I have some suggestions and recommendations.
- The employed mathematical model seems to be entirely an original contribution of the authors. Most of the equations are not referenced by a proper citation, turning difficult to check their correctness and originality. The equations found in the literature have to be referenced.
- Some Figures miss the measuring units for time and RUL.
- Which is the prediction error of the proposed model?
- The effect of increasing the batch of historical data of the machining process on the prediction error of RUL must be discussed.
- Reading the other papers on the connected subjects, e.g., [2], the prediction of the RUL seems more precise than the actual one. I advise the authors to validate their model with results obtained using other models from the literature.
Reviewer 3 Report
The reviewer comments of the paper «RUL Prediction of Cutting Tool Using Inverse Gaussian Process Model»- Reviewer
The authors presented an article «RUL Prediction of Cutting Tool Using Inverse Gaussian Process Model». However, there are several points in the article that require further explanation.
Comment 1:
Title.
Replace the abbreviation "RUL" with "remaining useful life".
Comment 2:
The abstract is well written.
However, add experiment cutting conditions, workpiece material and wear levels.
Please provide the main quantitative and qualitative research findings. What is the scientific novelty of the article? What is the practical value? What makes this approach different from other researchers? What is meant by precision? The article contains many formulas, but nothing is said about their physical meaning in the abstract. Everything should be briefly explained to the reader.
Comment 3:
The introduction is not written quite logically and clearly.
The main thing is what exactly do the authors mean by the concept of accuracy? Is it roughness? Form error? Linear dimension deviation?
All of these phenomena should be clearly discussed in the introduction. Otherwise, the "value" of the research data is low to say the least.
In this context, it is important to add articles in the introduction of authors Pimenov D.Y., Mikołajczyk, T. related to tool wear monitoring.
In addition, no one has previously solved the problem in the same setting as the authors? This should be clearly shown in the introduction. What are the "white" spots? That is, what has not a single researcher done before?
And most importantly, after reading the article to the end and the conclusion, it is still not clear how the remaining useful life is related to accuracy and what kind of accuracy indicator?
Comment 4:
- Methods
Are all the formulas in the article original? If not needed appropriate citations.
Show more clearly which of the formulas in section 2 are original? And most importantly, what distinguishes them from earlier studies?
Comment 5:
- Example study
3.1. Simulation
For what cutting conditions are figures 1, 2, 3, 4, 5, 6 obtained? For figure 5, you need to add units for each axis.
3.2. Experiment
Are all the figures in the article original? If not needed appropriate citations and publisher permissions.
For devices, software and machines used in research, indicate in parentheses (manufacturer, city, country).
The authors state about real turning experiments. However, the photo shows the milling process. Authors should clearly clarify all terminology in all sections of the article from title to conclusion. What instruments are used for roughness and tool wear measurements? Describe clearly and in detail the procedure and sequence of these measurements. At what point in time? During machining or during a stop? How many repetitions of measurements are used? On what surface? Flank face? What is the material of the workpiece? Give the chemical composition of the workpiece material in the table. What is the geometry of the tool? Dimensions, angles, etc.? What is the material of the cutting part? Are you using coolant or dry milling?
In Figure 8, show the experimental points?
The quality and resolution of all figures should be significantly improved.
Comment 6:
It will be useful to add a section of Nomenclature in which to sign all the physical quantities and abbreviations encountered in the article. There are many physical quantities in the text and such a section will help to find the description of the necessary element.
For example,
ap : Depth of cut (mm)
RUL : Remaining useful life
etc.
Comment 7:
The conclusions need to be improved.
What is the novelty of the article? What is the practical significance? What are the differences from previous works?
Please provide the main quantitative and qualitative research findings.
Conclusions should reflect the purpose of the article.
Use format.
∙ Conclusion 1
∙ Conclusion 2
∙ etc.
Comment 8:
Careful proofreading by a native English speaker is required.
The article has potential. However, in its current form, it is not suitable for publication in an international journal. And the main thing from the article is still not clear how tool wear and precision machining are still connected. These questions should be clearly explained to readers in all sections of the article. Authors should carefully study the comments and make improvements to the article step by step. All changes should be highlighted in color. After major changes can an article be considered for publication in the "Applied Sciences".
Round 2
Reviewer 1 Report
The manuscript has been amended according to the comments and it is ready to publish in the journal.
Reviewer 2 Report
The authors responded to all the questions and amended the paper as requested. I recommend this paper for publication.
Reviewer 3 Report
The authors have improved the article according to the comments. However, there are still a few important fixes:
1. Make sure that all physical quantities used in formulas 1-22 are included in the abbreviations section.
2. Include in references 22 and 29 all co-authors of articles.
After that, the article can be accepted for publication.
